



# Sensitive infrastructures and people with disabilities – Key issues when strengthening resilience in reconstruction

Alessa Truedinger[1], Joern Birkmann[1], Mark Fleischhauer[2], Celso Ferreira[3]

[1]Institute of Spatial and Regional Planning, University of Stuttgart, Stuttgart, 70569, Germany
[2]Department of Spatial Planning, TU Dortmund University, Dortmund, 44227, Germany
[3]Civil, Environmental, and Infrastructure Engineering Department, George Mason University, Fairfax, VA, 22032, USA

*Correspondence to*: Alessa Truedinger (alessa.truedinger@ireus.uni-stuttgart.de)

**Abstract.**

*The flood disaster of July 2021 claimed the lives of more than 220 people in Western and Central Europe - particularly severely affected was the Ahr Valley in Germany, where the floods caused at least 135 fatalities, damaged and destroyed more than 9,000 buildings, and caused billions of euros in damage. To prevent such a disaster from happening again, it is crucial not to simply rebuild, but to build up in a way that strengthens resilience to future events. Since time and money are often critical issues in the reconstruction process, it is important to focus on most vulnerable groups as well as critical and sensitive infrastructures, as these need particular attention and support for risk reduction and resilience building within the recovery process. The paper systematizes how critical and sensitive infrastructures are defined. It explores - based on the Ahr Valley flood disaster - how sensitive infrastructures can be identified and how they are treated and discussed in the recovery process. In addition, an easy-to-use framework for risk assessment and the subsequent selection of necessary measures is being developed. A detailed application of the framework assessment is carried out with regard to a school for children with disabilities that is located directly at the river Ahr.*

## 1 Introduction

Building back better and enhancing the resilience of communities and cities after disasters within the reconstruction process are globally important issues (United Nations Office for Disaster Risk Reduction, 2015). While much attention has been paid to the reconstruction process in general (United Nations Office for Disaster Risk Reduction, 2015) and housing reconstruction in particular (UN-Habitat and AXA, 2019), the issue of sensitive infrastructures and the specific protection needs of people with disabilities have often been overlooked and not sufficiently explored as well as considered (Kelman and Stough, 2015a; Ronoh et al., 2015; Ton et al., 2019; Global Facility for Disaster Reduction and Recovery and The World Bank Social Development Global Practice, 2020) - in Germany, as elsewhere, people with disabilities are mostly not sufficiently considered in existing disaster management concepts (Office of the Representative for the Interests of Persons with Disabilities and DRK Landesverband Baden-Württemberg, 2023). Yet, recent disasters underscore that these groups and facilities need more attention (Kelman and Stough, 2015b; Global Facility for Disaster Reduction and Recovery and The World Bank Social Development Global Practice, 2020).

The consideration of sensitive infrastructures with particularly vulnerable groups is less advanced as the discussion about critical infrastructures, such as electricity networks, in post-disaster reconstruction which has already been researched in various studies (Mulowayi et al., 2015; Sarker and Lester, 2019; Rouhanizadeh and Kermanshachi, 2020) - even with regard to the 2021 flood event (Koks et al., 2022), whereby this study at least addressed the areas of health and education, but no in-depth discussion was given to other sensitive infrastructures or facilities for people with disabilities. In addition, the topic of schools, for example, has so far mostly been considered in the context of earthquakes and earthquake-proof reconstruction or new construction (United Nations Centre for Regional Development, 2009; Asian Development Bank, 2015) - and less in the context of climate change-enhanced extreme events such as flooding. In light of changing risks due to those changing climatic hazards such as floods, heavy rainfall and heat, the vulnerability and the protection of sensitive infrastructure and its users



must receive a higher attention. In Germany, the new Federal Spatial Development Plan for Flood Protection (*Verordnung über die Raumordnung im Bund für einen länderübergreifenden Hochwasserschutz* - BRPHV) from 2021 underscores in its objectives, there is even a need to address the special requirements for protecting sensitive infrastructures and vulnerable groups and land-uses. While the formulation of the BRPHV supports the understanding that sensitive infrastructures and

vulnerable groups need particular attention, the nationwide document is and cannot be very precise. Furthermore, this plan is primarily addressing risk assessment before extreme events, while the concrete question on how to consider these aspects in reconstruction efforts is also new and an emerging issue. In addition, issues such as the different sensitivity of land uses and the varying vulnerability of user groups have not yet been central points in specific guidelines on reconstruction, and some guidelines in the field of water management and risk prevention often emphasize the risk cycle and the importance of property

protection measures - but do not adequately cover the role of sensitive and critical infrastructures.

Since both disaster and post-disaster reconstruction literature as well as official planning documents and regulations show a deficit with regard to the consideration and treatment of sensitive infrastructures, this paper develops and outlines a more systematic approach on how to deal with exposed sensitive infrastructures in reconstruction and recovery. In this regard our paper addresses the following research questions:


1. How to define sensitive infrastructures as opposed to critical infrastructures? How to better account for the special needs of vulnerable people in sensitive infrastructures - especially during reconstruction?
2. How to improve the consideration of sensitive infrastructures and most vulnerable groups in strategic assessment methods and in guidelines for recovery funding?
3. How to build back sensitive infrastructures better?

## 2 The flood disaster of July 2021 in the Ahr Valley

The Ahr Valley - a low mountain region with steep slopes and a narrow valley offering little space for settlement areas - encompasses a high exposure of infrastructures to floods and heavy precipitation as the event in 2021 dramatically revealed (Truedinger et al., 2023). Various schools, elderly homes, hospitals and care homes in the region, for example in the city of

Bad Neuenahr-Ahrweiler, are located close to the river - and thus have been adversely affected by the heavy-rainfall induced floods in 2021, which were caused by an upper-level trough that shifted eastward from the Atlantic Ocean to the southeast and encountered resistance from a quasi-stationary anticyclone positioned over northeastern Europe (Mohr et al., 2023). Heavy rainfall, on average about 75 mm across the Ahr catchment within a 24-hour period, resulted in severe flooding within the Ahr Valley, notably on July 14th and 15th, 2021 (Mohr et al., 2023).

Reconstruction approaches in the Ahr Valley and other affected regions in Germany mainly focus on the compensation of experienced losses and damages (Landesregierung Rheinland-Pfalz, 2021; Birkmann et al., 2023) - thus hampering resilience building and not sufficiently capturing the specific nature and needs of critical and sensitive infrastructures including the vulnerability of the persons using it (Birkmann et al., 2023). A drastic example of this so-called sensitive infrastructures is the "Lebenshilfehaus", a care home for people with disabilities, in the city of Sinzig. In the night of the flood event of 2021 twelve

people with disabilities lost their lives because the water rose enormously quickly and the management of the house and the local disaster protection units were unable to save these persons (SWR, 2021). In this context, it is important to mention that the city of Sinzig is located downstream, which means that the flood had already destroyed places up-stream in the afternoon and evening of the same day.

The challenges on how to better prepare sensitive infrastructures with highly vulnerable population groups is not limited to

this single case, but is an important emerging issue in the reconstruction process that is unsolved up to now. The Levana School in Bad Neuenahr-Ahrweiler, that is discussed in this paper in-depth, is another example. The school is a facility for children





with special needs. Since the flood of 2021 occurred during the evening and night, fortunately no students were in the school. At another time of the day, however, there could have been fatalities as well - and the inventory and the school building itself suffered severe damage in 2021 (Himmelrath, 2022). Next to direct damages also the losses of school and teaching time need

to be considered as an important secondary damage.

In this regard, the 2021 flood disaster in the Ahr Valley in Germany and the subsequent reconstruction process is a good example and a powerful case study to explore our research questions and the importance of vulnerability and sensitivity in building resilience and to find concrete options for building back better sensitive infrastructures.

### 3 Critical versus sensitive infrastructures in risk management and reconstruction

While the protection of critical infrastructures has received an increasing attention - in Germany, e.g. within the Federal Spatial Planning Act (*Raumordnungsgesetz* - ROG) -, the consideration of specific protection needs of sensitive infrastructures, such as schools, elderly homes or kindergartens is still limited and less advanced. The flood disaster of July 2021 significantly revealed the lack of preparedness of sensitive infrastructures in Germany - as it is also the case in other countries, e.g. the United States of America. The event of Hurricane Ian that impacted the Florida coast in 2022 showed that critical public

healthcare inadequacies disproportionately affected the older adult population and resulted in fatalities after Hurricane Ian (Bushong and Welch, 2023). Another study investigating if disabled individuals were disproportionately impacted by Hurricane Harvey in Texas found that the overall extent of Harvey-induced flooding was significantly greater in areas where a higher proportion of disabled residents lived (Chakraborty et al., 2019).

Without the functioning and rapid (re-)construction of critical and sensitive infrastructures, communities cannot recover and

find a way to build resilience. So far, reconstruction and recovery often focus mainly on the physical rebuilding of these infrastructures, rather than on the loss of functions - such as schooling. In this respect, it is not just about rebuilding the physical structure, but resilience also means that people and children have trust and feel safe in the place where they are accommodated and can reliably use its functions.

Before going into the assessment of how sensitive and critical infrastructures are at risk and how they are treated within the

recovery and reconstruction process, we present core definitions based on a literature review and the analysis of selected laws, directives and official documents in Germany/EU as well as the US to better differentiate the two terms and outline the current legislation with regard to these infrastructures.

### 3.1 Definitions

**Critical infrastructures:** As already stated, much attention has been paid to the concept of critical infrastructures in the last

years. As a result, slightly different definitions can be found, which nevertheless essentially contain the same meaning. Critical infrastructures are objects, installations, networks, systems, facilities or organizations providing services that are vital for the functioning of the community or society and whose failure or impairment would lead to serious consequences, e.g. disruptions to public safety or shortages of essential goods (Federal Office for Information Security, n.d; United Nations Office for Disaster Risk Reduction, n.d; Stewart et al., 2009). Depending on the source of information, this includes various sectors (Filiol and

Gallais, 2014). The following sectors are often mentioned: energy, water supply and disposal, transportation and traffic, finance, healthcare, government and public administration, information and communication technology, media and culture, food as well as waste disposal (Federal Office for Information Security, n.d; Stewart et al., 2009). In the European Critical Entities Directive (CER), for example, space is also listed as a sector (European Parliament and Council, 2022). Outstanding in critical infrastructures are the interconnectivities and (inter-)dependencies which can lead to so-called domino and cascade

effects in the event of an impairment or failure of one or several critical infrastructures or critical infrastructure components (Hellström, 2007).



**Sensitive infrastructures:** Sensitive infrastructures with a particularly vulnerable population or user group have not yet been the focus of attention. In fact, there is not even a standardized definition in Germany, as is the case with critical infrastructures (Hartz et al., 2020). Furthermore, in the US, for example, the concept of critical infrastructure is defined more broadly by the

Federal Emergency Agency (FEMA) than is the case in Europe and Germany - here the concept of critical infrastructure includes not only facilities that are vital for the population, but also, for example, facilities that are essential for the protection of specific population groups (Federal Emergency and Management Agency, 2007). Greiving et al. (2023) however, provide a proposal for the term of sensitive infrastructures - they define it as infrastructures that are used by groups of people who require assistance from third parties in the event of an incident. This is partly in line with a description of the term in a

publication by the German Federal Institute for Research on Building, Urban Affairs and Spatial Development, whereby the term in this publication is not limited exclusively to facilities whose users are dependent on assistance in the event of an incident, but also includes other infrastructures that can be of great importance to the community and whose users or uses are very vulnerable, but whose failure does not necessarily lead to significant supply shortages or threats to public safety (Hartz et al., 2020). In addition to kindergartens, school facilities for children with disabilities or retirement and care homes, these

can also include large stables in agricultural production, for instance. Nevertheless, it is important to emphasize that for people whose mobility or perception is limited - be it due to physical or mental disabilities, due to advanced pregnancy, due to illness, etc. - flooding is definitely a matter of life and death. Consequently, we consider infrastructures that are utilized by users who require assistance from third parties or special technology in the event of an incident, i.e., primarily people with limited mobility or perception or ability to express themselves, to be sensitive.

The difference in a nutshell: While the focus when considering critical infrastructures is usually on the service they provide to the community, the focus when considering sensitive infrastructures is on the user group, i.e. the living beings that use the infrastructure or regularly spend time in it. It should be noted that there is an overlap - for example, hospitals can be assigned to both terms respectively concepts. Hence, another concept is introduced - that of protection worthiness (Greiving et al., 2023).

**Protection worthiness:** Protection worthiness is a political-normative concept, therefore a broad discourse and a politically legitimized system of objectives is necessary to determine which infrastructures belong to this concept (Hartz et al., 2020) - whereby, depending on the political norm, critical as well as sensitive, endangering and particularly meaningful infrastructures can be included (see Table 1 and (Greiving et al., 2023)). If the concept is applied in spatial planning, this means that certain land uses and spatial functions are given greater weight in the question of protection - with corresponding consequences in the

context of a balancing process (Hartz et al., 2020; Greiving, 2023).

**Table 1: Excerpt of possible reasons for special protection needs of different types of infrastructures, adapted from (Greiving et al., 2023).**

| Infrastructure type | Examples | Legal basis and official documents/ requirements (Germany/EU and USA) | Reasons for protection worthiness |
|---|---|---|---|
| Sensitive infrastructures | - Kindergartens<br>- Schools<br>- Senior citizens and care facilities | - In parts, e.g. State law on fire protection, general assistance and civil protection (*Brand- und Katastrophenschutzgesetz –* LBKG) of Rhineland-Palatinate<br>- US Risk Management Series Design Guide for Improving School Safety in Earthquakes, Floods, and High Winds Design Guides (Federal Emergency and Management Agency, 2004) | Avoidance of personal injury to groups of people who require assistance from third parties in the event of an incident |



| Critical infrastructures whose failure or impairment results in lasting disruptions to the overall system/to the society | - Supply networks (gas, water, electricity, telecommunications) - Transportation networks | - Directive (EU) 2022/2557 of the European Parliament and of the Council of 14 December 2022 on the resilience of critical entities and repealing Council Directive 2008/114/EC - BSI Critical Services Ordinance (*Verordnung zur Bestimmung Kritischer Infrastrukturen nach dem BSI-Gesetz* - BSI-KritisV) - US Risk Management Series Design Guide for Improving Critical Facility Safety from Flooding and High Winds (Federal Emergency and Management Agency, 2007) | -Avoidance of disruptions and shortages - Avoidance of loss of function outside exposed areas and in other infrastructure sectors (so-called "domino and cascade effects") - In the US, the term of critical infrastructures is used much more broadly by the Federal Emergency and Management Agency (FEMA), so that "sensitive infrastructures" are also included here, as FEMA defines that critical facilities comprise all public and private facilities deemed by a community to be essential for the delivery of vital services, protection of special populations, and the provision of other services of importance for that community. |

Despite the different laws and publications that underscore the importance of critical and sensitive infrastructures in the context of extreme events and disasters, there are no standard procedures in Germany for identifying and assessing or evaluating critical and sensitive infrastructures in local and regional planning in the context of extreme events. Also, the formulation of protection goals for these types of infrastructures and their users is still absent.

### 3.2 Legal basis

In addition to the European Flood Risk Management Directive (FRMD), the BRPHV, which came into force in September 2021, the Spatial Planning Act, the Building Code (*Baugesetzbuch* - BauGB) and the Federal Water Act (*Gesetz zur Ordnung des Wasserhaushalts* - WHG) set out legal principles for flood risk management and flood prevention in Germany. With regard to spatial planning, the general "precautionary principle" within the Spatial Planning Act (Sec. 1 para. 1 No. 2 ROG) and the principle set out in Sec. 2 para. 2 No. 3 sentence 4 ROG, which states that the "protection of critical infrastructure shall be

taken into account", should be particularly emphasized. The Federal Building Code also specifies in Sec. 1 para. 6 No. 12 BauGB that the "concerns of coastal or flood protection and flood prevention, in particular the avoidance and reduction of flood damage" are to be taken into account in settlement and infrastructure development as well as in the preparation of urban land use plans. However, in this case, urban land use planning is aligned with floodplains (Sec. 76 para. 1 WHG), within which there are far-reaching building bans in accordance with Sec. 78 para. 1 WHG. In this context, the load case that serves as the

basis for determining the floodplains is merely the design event that occurs statistically every 100 years (HQ-100) - no distinction is made between different hazard intensities, sensitivities and necessity of special protection due to different levels of vulnerability (Greiving, 2021). So far, climate change is not considered in this designation either - only in some federal



states are changes due to climate change considered in the assessment of technical facilities (KLIWA Klimaveränderung und Wasserwirtschaft, n.d.), but not in terms of spatial planning and the designation of flood-prone areas.

Since its publication in 2021, the BRPHV calls for the consideration of risk-based planning approaches in spatial and urban planning. The BRPHV and its principles have to be taken into account by all public planning authorities and thus, next to the often considered aspect of the probability of occurrence, the Federal Plan also requires authorities to take into account the flow velocity and flooding depth as well as the sensitivity of exposed land-uses and infrastructures and respective protection requirements. However, this only has to be done on the basis of data available from public bodies. Therefore, the

implementation of these enhanced principles is still lacking or solely done in part, since the data available at public agencies is not sufficient to systematically consider these aspects. Particularly, data on the sensitivity of different land-uses and data regarding the vulnerability of different infrastructures and their users as well as regarding protection requirements is often not available for planning authorities and emergency response units (Schulze et al., 2019). Also, further specifications with regard to the question of how to account for different flow velocities and flooding depths levels are still lacking. According to Principle

II.2.2 (G), the BRPHV also requires that in floodplains according to Sec. 76 para. 1 WHG, the replanning and reconstruction of "existing settlements or settlement structures [should be pursued] in a medium-term timeframe, insofar as the spatial situation in the affected municipalities and the law on the protection of historical monuments permit and insofar as this is more cost-effective in the long term from an economic point of view than land or property protection".

With regard to critical infrastructures and IT security, the BSI Critical Services Ordinance (*Verordnung zur Bestimmung*

*Kritischer Infrastrukturen nach dem BSI-Gesetz* - BSI-KritisV) defines critical services as well as thresholds above which the reporting and verification obligations of the Law on the Federal Office for Information Security (*Gesetz über das Bundesamt für Sicherheit in der Informationstechnik (BSI)* - BSIG) apply, although the relatively high thresholds of 500,000 people supplied often mean that locally and regionally significant critical services are not covered. Moreover, the European Union's Directive on Critical Entities Resilience (CER Directive) adopted in January 2023 contains specifications for identifying and

strengthening the resilience of critical infrastructures - which is to be anchored in German law as part of the planned "Critical Infrastructure Umbrella Law" (revised draft available as of December 2023). This planned law is intended to systematically, comprehensively and uniformly identify critical infrastructures at federal level, recognize risks and disruptions to the overall system, increase protection levels through binding minimum requirements and create an institutional framework for cooperation (Bundesministerium des Innern und für Heimat, 2023). Nevertheless, only those critical infrastructures that are

essential for the overall supply of Germany and serve at least 500,000 people must take mandatory resilience measures (Bundesministerium des Innern und für Heimat, 2023). For sensitive infrastructures, on the other hand, there are hardly any specific legal requirements or binding protection targets in Germany - regardless of which level is being looked at. According to the Building Code, old people's as well as care and nursing homes are considered unregulated special buildings ("ungeregelte Sonderbauten"), meaning that there are no specific structural, systems engineering or organizational

requirements for this type of building use. So, there are neither legal requirements in the context of risk management, nor methods for identification, nor guidelines for resilient sensitive infrastructures. The only document in Germany to date that also addresses these infrastructures with their particular need for protection is the aforementioned BRPHV, which explicitly calls for the different sensitivities and protection worthiness to be considered in spatial planning in the sense of a risk-based approach, albeit it remains rather vague overall. In other countries, attention is paid to some of those sensitive infrastructures,

but often in the context of earthquakes, e.g. earthquake-proof schools (United Nations Centre for Regional Development, 2009) - and thus less in the context of flooding or explicit reconstruction processes.

So far, sensitive infrastructures with particularly vulnerable groups have not been adequately addressed. In many cases, the focus is more on a technical understanding of processes that are particularly critical and important for the public, like electricity and communication infrastructure, sometimes hospitals, but there is a lack of considering other important infrastructure and

groups of people with high vulnerability that require particular attention, especially during reconstruction.



### 3.3 The impacts of the Ahr flood 2021 on critical and sensitive infrastructures

The devastating floods in summer 2021 washed away and destroyed lots of critical and sensitive infrastructures, since many of them were directly located close to the river. A total of 17 schools were hit particularly hard by the flood (Die Landesregierung Rheinland-Pfalz, 2022), so that no classes could be held there after the flood for several weeks and months.
In addition to schools, 42 kindergartens and daycare centers were affected in the county of Ahrweiler (Bundesministerium des Innern und für Heimat and Bundesministerium der Finanzen, 2022). In Bad Neuenahr-Ahrweiler, the largest town in the Ahr Valley, eight kindergartens and daycare centers were damaged. Since most of the children attending these facilities are six years old and younger, they are seldom able to get to safety on their own in the event of flooding, simply because of their physical and mental condition. Also, an integrative daycare center in the direct vicinity of the Ahr was affected that accommodates children with disabilities as well as infants and toddlers. Many other sensitive infrastructures were also affected - such as care facilities, 15 of which had to be evacuated in the county of Ahrweiler (Bundesministerium des Innern und für Heimat and Bundesministerium der Finanzen, 2022). The aforementioned case of the "Lebenshilfe" care home in Sinzig, where 12 residents drowned, was particularly tragic. Another illustrative example is the Levana School that was severely affected by the floods in 2021 and accommodates children with physical and mental disabilities. Solely the direct impact of the flood caused millions of euros of damage.

Furthermore, various critical infrastructures were affected, such as the majority of the fire stations or four hospitals and specialized clinics. Among them was an emergency hospital, which, however, was able to resume business successively just one month after the event (Marienhaus Klinikum im Kreis Ahrweiler, 2021). In addition, tens of thousands of households were temporarily cut off from the power supply - and communications, gas and rail networks, roads and bridges, as well as water supply and wastewater disposal were (Die Landesregierung Rheinland-Pfalz, 2022) and still are affected in some cases.

### 4 Methods

To explore the different aspects and challenges of strengthening protection and resilience - especially within the reconstruction process - of sensitive infrastructures, we have undertaken a mixed-method approach in order to triangulate methods and data. Our mixed-methods approach includes, in particular, a literature and document analysis as well as expert interviews, workshops and discussions, on the basis of which the assessment method was developed. Afterwards, a detailed case study was used to test the applicability of the assessment method, including other methods such as GIS analyses as well as observations and assessments of the flood impacts, the location and the construction of the building. The results of a quantitative household survey were also integrated in order to include the views of another stakeholder group in addition to experts such as school authorities, school management and municipal stakeholders. All the methods used are listed in Table 2, along with the objectives and justification for the use of each method and the data obtained, where this can be reasonably specified. The various methods are described in more detail in the following subsections.

**Table 2:** Presentation of the methods used, the objectives and justification for the use of the methods and the data obtained (to be stated where appropriate).

| Methods | Goals / Justification for use | Received / Analyzed data, contents and meetings |
|---|---|---|
| Literature and document analysis | - Obtaining knowledge about the current state of research, the current state of reconstruction and the definitions of various terms | - Scientific publications related to critical infrastructure, sensitive infrastructure and people with disabilities in connection with reconstruction after natural disasters<br>- Guidelines for reconstruction funding<br>- Legally binding spatial planning documents (in Germany)<br>- Laws and guidelines with regard to spatial planning, flooding and critical infrastructures |




| | | - Official reports on the flood disaster of July 2021, the reconstruction process and the findings of committees of inquiry, etc.<br>- Documents provided by authorities and agencies (e.g., evacuation plans) |
|---|---|---|
| Expert interviews | - Obtaining detailed expert knowledge | - On-site visit and conversation with the managing director of the Construction and Development Company Bad Neuenahr-Ahrweiler (Aufbau- und Entwicklungsgesellschaft Bad Neuenahr-Ahrweiler) on 05.07.2023 (Bad Neuenahr-Ahrweiler)<br>- Semi-structured interview with a school staff member of Levana School on 12.06.2023 (online)<br>- Unstructured interview with a former school staff member of Levana School on 25.04.2023 (by telephone)<br>- Several brief telephone conversations to clarify specific questions also took place, for example, with an expert from SGD Nord (Upper State Authority of Rhineland-Palatinate) |
| Expert workshops and expert discussions | - Obtaining detailed expert knowledge<br>- Assessment/Verification of the new findings by experts within their (guided) discussions | Participation and minutes of several meetings in the KAHR project context, e.g.<br>- With the county administration and the "Owner-operated Municipal Enterprise Schools and Facility Management" (Eigenbetrieb Schulen und Gebäudemanagement) on 05.04.2023 and 22.05.2023 (online)<br>- With state, regional and county planning on risk-based spatial planning, e.g. with regard to schools on 07.02.2023 (online) |
| Household survey | - Obtaining a comprehensive picture of the residents' views on sensitive infrastructures, their worthiness of protection and possible measures | - Survey data |


A case study was then used to test the assessment method developed using the aforementioned methods. In this test of real-life applicability, maps, flood impacts and various documents and personal interviews were also analyzed and included.

### 4.1 Qualitative literature and document analysis

As the reconstruction process in the Ahr Valley and the other regions is still in progress and the event itself happened only a few years ago, there are not yet many scientific publications relating to the Ahr flood of 2021. Nevertheless, scientific publications were included in the literature and document analysis where appropriate - in part also in relation to the overarching topic of critical and sensitive infrastructures as well as people with disabilities in the context of flooding and reconstruction processes. In addition to a number of scientific publications, also public documents such as reports from various authorities and committees of inquiry, funding guidelines from the federal and state government, minutes of meetings and resolutions 260 from various bodies as well as planning and legislative documents were used.

### 4.2 Expert interviews, workshops and discussions

Expert interviews, workshops and discussions were conducted in various settings to expand knowledge by obtaining expert knowledge and to assess and verify the scientific findings. For example, a semi-structured interview was conducted with a



school staff member of the Levana School, and a telephone interview was also held with a former school staff member to assess the location and the challenges posed by the evacuation issue. A site visit and discussion were conducted with the managing director of the construction and development company, whereby the city of Bad Neuenahr-Ahrweiler is, e.g., the responsible authority for the inclusive kindergarten "St. Hildegard", which is located in close proximity to the original location of the Levana School. In addition, several expert discussions took place with the county administration and the "Owner-operated Municipal Enterprise Schools and Facility Management" (Eigenbetrieb Schulen und Gebäudemanagement), where

the limits and possibilities of funding and relocation were also discussed. A more general meeting was held with state, regional and district planning on risk-based spatial planning, e.g. with regard to schools.

### 4.3 Quantitative household survey

A quantitative household survey with a standardized questionnaire was also carried out as part of the KAHR project to supplement the qualitative results. The household survey was prepared jointly by the Institute of Spatial and Regional Planning

at the University of Stuttgart and the Institute of Environmental Science and Geography at the University of Potsdam and distributed with the help of the county of Ahrweiler. For this purpose, around 5250 people from the group of people who had applied for emergency aid after the Ahr flood were contacted in mid-June 2022 and asked to take part in the survey with around 30-40 letters being returned as undeliverable. Between June and August 2022, 516 people, and thus 9.9% of those contacted, took part in the survey - the majority of them online. Nevertheless, the option of a paper questionnaire was also offered so that

older people in particular could be reached. The questionnaire was created using the EvaSys Survey software and contained both multiple choice and single choice as well as dichotomous and Likert scale questions. In addition to the mostly closed questions, a few free-text questions were also included. To safeguard confidentiality and prevent random responses, participants were given the choice to opt for a "no response" option for each question. However, it was mandatory to explicitly indicate this option for the questionnaire to proceed to the subsequent stages of processing. Therefore, it took about 30-45

minutes to complete the questionnaire. Prior to the actual household survey, the questionnaire underwent a preliminary testing phase, during which it was reviewed by several on-site project staff members who were directly impacted. The questions ranged thematically from the 2021 flood event to the coping and reconstruction process to prevention and preparedness. The survey was analyzed using the software IBM SPSS Statistics, version 28.0.0.0 (190) and version 29.0.0.0 (241). More detailed information about the household survey can also be found in (Truedinger et al., 2023).

### 5 New risk assessment framework for sensitive infrastructure

Based on the literature and document analysis as well as the expert interviews, workshops and discussions, a more systematic approach to assess the risk of sensitive infrastructures, with a particular emphasis on both the structure of the building and its function as well as the vulnerability level of the user group and their capacity to adapt and cope with extreme events - which is in our case flooding - is shown below. In addition, the results of the applicability test and the demonstration of the evaluation

method are presented using a detailed case study.

Thus, results presented and discussed in the following are not only relevant for the individual case study, but also provide a basis for a systematic assessment approach to better account for risk reduction and resilience building of sensitive infrastructures in reconstruction processes - and also in completely new planning processes, as reconstruction after disasters can also be seen as a new construction.

### 5.1 Systematic approach to determining the risk a sensitive infrastructure is facing

The following systematic approach to determining the risk, that a sensitive infrastructure is facing with regard to flooding, is based on our own considerations, analyses and discussions in the KAHR research project and builds on the current literature.



Findings from intensive discussions and interviews with different stakeholders (see Table 2) as well as from literature und document analysis have also been included. After application of the systematic risk determination approach, measures for

reconstruction and new construction can be derived (see Chapter 6.1).

The diagram depicted in Fig. 1 illustrates the systematic framework, Table 3 contains the underlying questions.

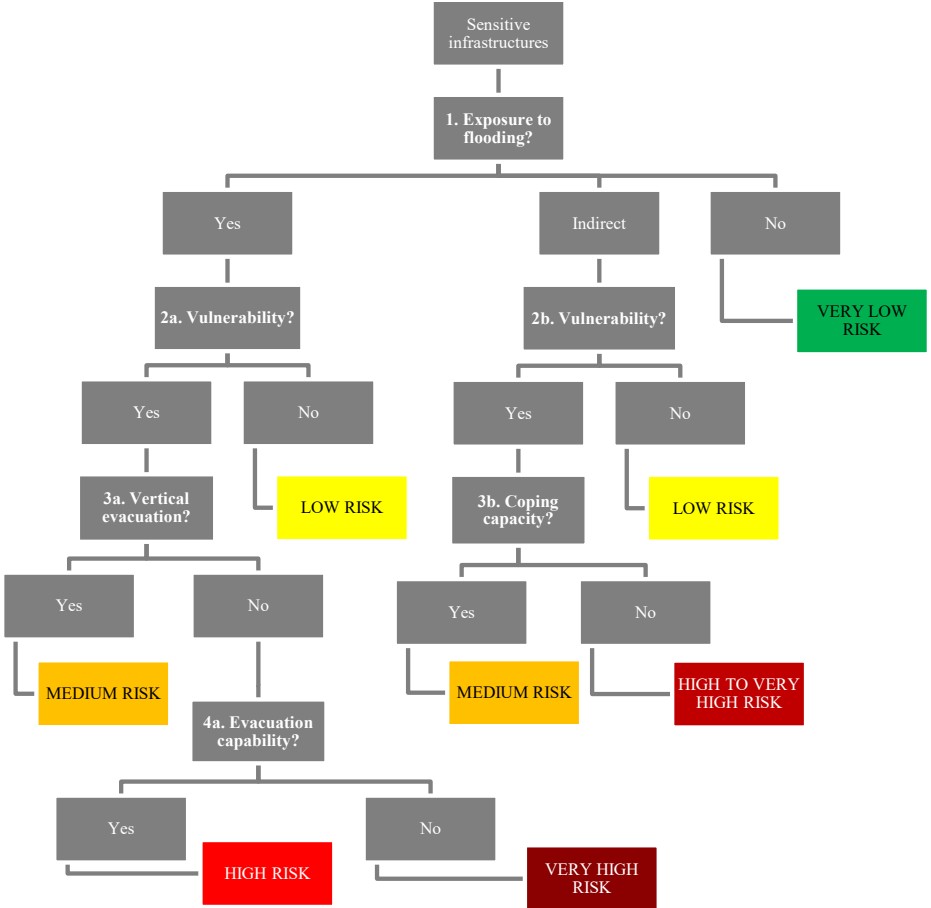

**Figure 1: Systematic framework for determining the risk that a sensitive infrastructure is facing (own figure).**


The diagram (Fig. 1) illustrates the logic and simplicity of the systematic approach. The questions behind the respective points 1-4, which should be used for assessment, are contained in the following Table 3 as well as possible sources of information and important remarks.

**Table 3: Important questions and notes on the risk assessment according to diagram shown in Fig. 1 (own table).**

| Point | Questions | Sources of information | Important remarks |
|---|---|---|---|
| 1. Exposure? | - Is the infrastructure/ Are the users exposed to flooding - not only to 100 yr events/ HQ-100, but also to extreme events/ HQ-extreme? | - Flood hazard maps<br>- Heavy rain/ Flash flood hazard maps | - Flood hazard maps could vary from federal state to federal state in Germany as, e.g., the definition of HQ-extreme varies - sometimes |



| | | | |
|---|---|---|---|
| | - Is the infrastructure/ Are the users exposed to heavy rain flooding / pluvial floods / flash floods?<br>- Is there any indirect exposure of the infrastructure, e.g. power outage or failure of the water supply possible due to supplying power or water infrastructure in exposed areas (HQ-extreme/ areas prone to heavy rainfall/ pluvial floods/ flash floods)? | - Studies/ modeling by research, engineering offices, etc.<br>- Operators of the possibly exposed critical infrastructures on which the vital dependency exists | HQ-200, sometimes HQ-1000, sometimes HQ-100 x 1.2 etc.<br>- Up to now, only the HQ-100 has been used as a basis for assessment in Germany; we recommend (also due to climate change) at least the HQ-extreme as a basis for assessment<br>- In the US, FEMA is, e.g., already transitioning from the 100 yr flood event to a risk-based approach<br>- Heavy rain/Flash flood hazard maps are partly not publicly available and often vary greatly (quality, method, level of detail…) and sometimes cities etc. have also produced individual maps. |
| 2a. Vulnerability? | Are the users particularly vulnerable? I.e.<br>- Do they have limited mobility, e.g. due to disabilities, pregnancy, age (under 6 or over 64)?<br>- Do they have limited perception and/or limited ability to articulate, e.g. due to disabilities or diseases? | - Information may be available, e.g., from the health department or head of the institution or from medical files (data protection!)<br>- Assessments of the personnel<br>- Assessments of the users itself | - If some of the users rely on critical services (e.g. if they are dependent on power-driven devices), one should additionally check the branch of indirect exposure as well and, in the case of a climate-resilient construction at the same location, appropriate measures should be taken with regard to these indirect impacts. |
| 2b. Vulnerability? | - Is the functioning of the fundamental infrastructure services (from the user's perspective) dependent on those other affected (critical) infrastructures?<br>- In the event of an indirect impact in the shape of a power outage: Are the users dependent on power-driven devices? | - Operators of the infrastructure(s)<br>- Assessments and examinations of experts | |
| 3a. Vertical evacuation? | - Is vertical evacuation to a higher floor - which is high enough even under extreme event conditions - possible in a very short time?<br>- Is it also possible to evacuate from this higher floor later on?<br>- Does vertical evacuation require personnel or special equipment?<br>- Can the users be informed of the need for vertical evacuation at any time and in an accessible manner? | - Building plans<br>- Emergency plans<br>- Test runs<br>- Experiences<br>- Assessments of the personnel<br>- Assessments of the users itself | - Definitely keep in mind: if there is any threat of heavy rain (which can occur suddenly and anywhere) and flash floods, there might be little to no warning time!<br>- In Sinzig, for example, there was only one night watch, which could not evacuate all the residents at the same time and in a timely manner |
| 3b. Coping capacity? | - Does the infrastructure have an instant emergency supply for 24h?<br>- Does the infrastructure have an instant emergency supply for 72h? | - Emergency plans<br>- Evacuation plans<br>- Previous experiences<br>- Assessments of experts | - If users are dependent on power-driven devices, these devices must be able to be supplied directly by external batteries or an emergency |



| | | | |
|---|---|---|---|
| | - Does the infrastructure have feed-in options (esp. when emergency supply only for 24h)?<br>- Is the evacuation capability so high that the infrastructure can be safely evacuated before any damage is caused by the failure (e.g. if all persons can be evacuated without special equipment and with the available personnel to a suitable location before any significant damage)? | | power supply in the event of a power failure |
| 4a. Evacuation capability? | - Is there sufficient advance warning time to carry out an evacuation, which can be personnel-, organizational-, time- and material-intensive?<br>To consider (among other things):<br>- Can the persons be evacuated not only from the building, but also from the flooded area?<br>- Is no additional personnel required for this evacuation? I.e. is the personnel available on a daily basis sufficient to evacuate all persons from the flooded area?<br>- Is no additional material required for this evacuation, e.g. special vehicles?<br>- Is a usable escape route available that is not prematurely flooded?<br>- Is the access route accessible with the existing vehicles and/or passable in the event of an incident?<br>- Is there a safe place nearby to evacuate to?<br>- Are the users still sane in extreme situations?<br>- Can the users be informed of the need for evacuation at any time and in an accessible manner? | - Evacuation plans<br>- Building plans<br>- Previous experiences (e.g. interviews with those responsible/ affected who have already had experience) or experiences from exercises<br>- Type of disabilities and impairments (may be available, e.g., from the health department or head of the institution)<br>- Assessments of the personnel<br>- Assessments of the users itself | - In contrast to the case of fire (which is usually practiced regularly and for which evacuation plans and routes out of the building are available), evacuation in the event of flooding is rarely considered → in this case it is essential to ensure that a usable route out of the flooded area is also available (even shallow water depths and flow velocities can be insurmountable for persons with limited mobility or normal vehicles!)<br>- Definitely check if there is any threat of heavy rain (can occur suddenly and anywhere) and flash floods → little to no warning time! |

## 5.2 Application of the assessment framework to the Levana School case study

In the following, the assessment framework is now applied to the case study of the Levana School to test the assessment method developed previously using a real-life and transferable example. By applying it, it will become clear how it is applied and what needs to be considered. In addition, various aspects of our assessment framework are also explained in more detail.

Depending on the expected risk, appropriate options for action can then be taken (see Chapter 6.1).

The Levana School (see Fig. 2), which is run by the county of Ahrweiler and situated in the city of Bad Neuenahr-Ahrweiler, is a school with the focus on holistic and physical development and was attended by a total of 92 students in the school year 2022/2023, whereby all pupils of the school have a mental disability. Of these, 30 students had the special focus on physical development in 22/23 so they have both mental and physical disabilities. In the future, however, due to the division of students

between the Burgweg School in Burgbrohl and the Levana School, it is to be expected that more students who also have physical disabilities will be enrolled at the Levana School, since this focus cannot be served by the Burgweg School.





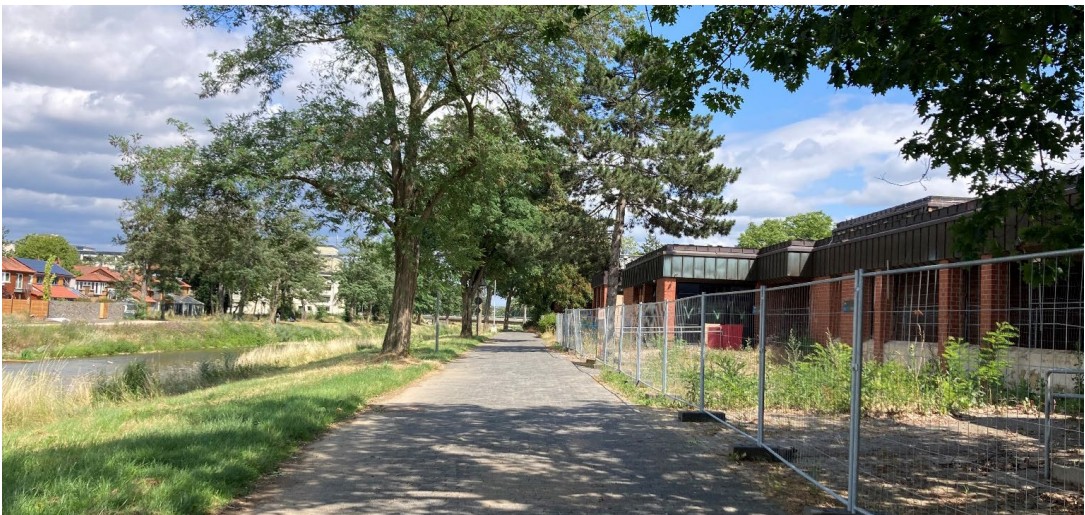

**Figure 2:** The school complex "Levana School and Don Bosco School" (right) is located in the immediate vicinity of the Ahr (left) and was flooded to a height of approx. 2 m in 2021, with the Levana School in particular accommodating highly vulnerable pupils (Photo: Truedinger, 330 2023).

Therefore, the Levana School is a particularly sensitive and protection-worthy infrastructure, which, incidentally, is also very exposed to flooding and heavy rain, and thus lends itself as a case study for theory testing. The transferability to many other special schools - especially those with the special need areas "mental development" plus "physical and motor development" - is given, as the design of schools for children with physical disabilities is usually very similar due to accessibility and as the 335 composition of pupils always varies from year to year - even in the Levana School itself.

### 5.2.1 Exposure

Within our assessment framework, exposure is the first, decisive factor in determining the risk and the subsequent choice of measures. If an infrastructure is neither exposed to riverine nor to pluvial flooding - although the latter can rarely be completely ruled out, as heavy rainfall events can occur ubiquitously (Weißer et al., 2020) - the only question that ultimately arises with 340 regard to the issue of flooding is whether the infrastructure could still be indirectly affected. This may be the case, for example, if the power supply to the infrastructure is potentially at risk of flooding. In instances of specific dependency of infrastructure or its users on critical services such as electricity, indirect impacts can yield fatal consequences. This is exemplified in the context of care facilities where individuals sometimes depend on life-sustaining equipment like oxygen devices. Direct exposure can be determined, for instance, through flood and heavy rain hazard maps, as well as past experiences. Additional 345 data is required for assessing indirect exposure, which may not always be readily available. Relevant service providers, such as utilities, might offer assistance in acquiring such data. Furthermore, hazard maps can be used to assess potential impacts on access roads - in the event of road inundation, evacuation and the arrival of emergency services could be impeded.

Currently, due to the flood of 2021, the Levana School is housed in containers on Schützenstraße in Bad Neuenahr-Ahrweiler, although the original location is next to the Don Bosco School (special school with a focus on learning and language) on St.- 350 Pius-Straße in the immediate vicinity of the river Ahr (see Fig. 3), with the main exit - marked with a black arrow - pointing directly towards the Ahr. The original building, which could be restored with the help of state subsidies, is exposed in various aspects. On the one hand, it is situated within the current HQ-100 zone, where, in certain areas, water levels would reach 1 - 2 m during a hundred-year flood event (see Fig. 3). According to the SGD Nord authority (*Struktur- und Genehmigungsdirektion Nord* – SGD Nord), the water at the main entrance to the building is around 60 cm high at the new HQ-100, which has been 355 recalculated and reclassified by the State Office for the Environment Rhineland-Palatinate (*Landesamt für Umwelt Rheinland-*



*Pfalz* - LfU RP) after the 2021 flood. Additionally, parts of the only access road to the building (see Fig. 3, marked with a grey arrow) are flooded even higher. At the lowest terrain point of the road, the water is, according to the SGD Nord, 0.99 m high during an HQ-100 event. The flood hazard map even shows a water height of at least one meter (see Fig. 3). Even at a gauge level of 410 cm in Altenahr, part of the access road and the collection point of the Levana School is already flooded (SGD Nord, n.d.), whereby such a level - according to the old calculations - occurs on a statistical average slightly less frequently than every 20 years (Landesamt für Umwelt Rheinland-Pfalz, n.d.). As a result of climate change, such events will occur more frequently from a statistical point of view.

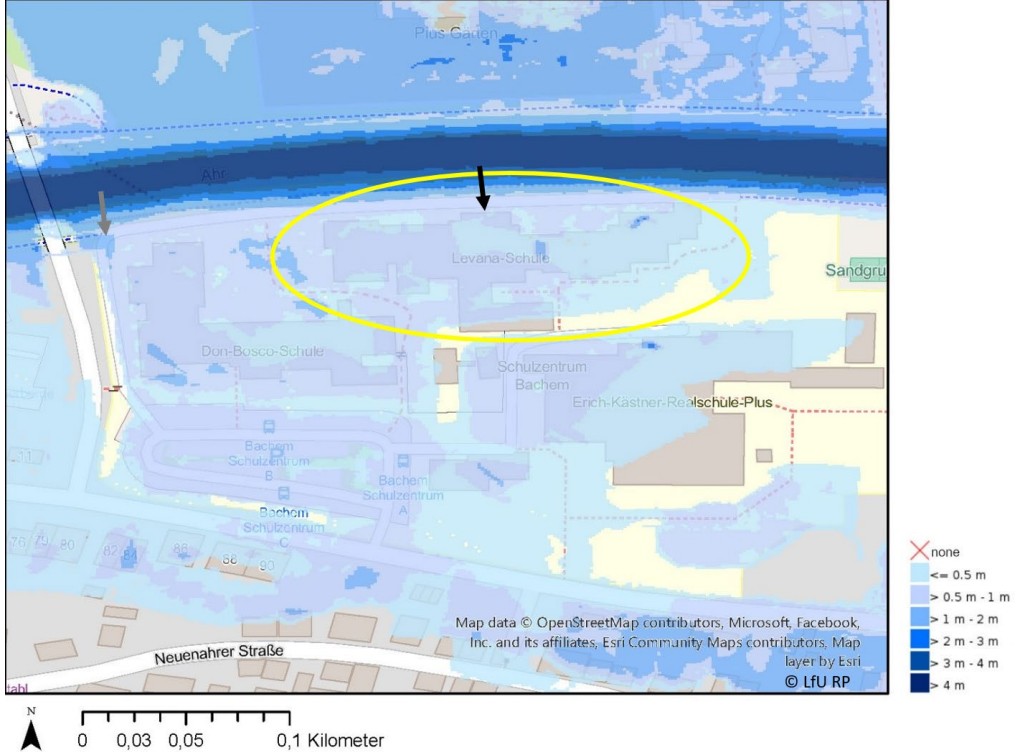

**Figure 3:** Depiction of the HQ-100 floodplains, revised by the State Office for the Environment Rhineland-Palatinate (Landesamt für Umwelt Rheinland-Pfalz (LfU RP)) after the 2021 flood, in the vicinity of the Levana School (Hochwassergefahrenkarte Rheinland-Pfalz, © LfU, https://wasserportal.rlp-umwelt.de/kartendienste).

The Levana School is also exposed to heavy rain events and flash floods. According to the latest calculations from the State Office for the Environment Rhineland-Palatinate (see Fig. 4 and Fig. 5), the flow velocity of the water along the access road is up to 1 m/s with a water depth of up to 30 cm, already in the event of exceptionally heavy rainfall (SRI7, which corresponds to a hundred-year event). On the part of the building facing away from the Ahr, the calculated water levels in this scenario are even up to 1 m with flow velocities of mostly 0 to 0.5 m/s, in a few places even up to 1 m/s. This means that even a "merely" exceptional heavy rainfall event results in partly high exposure - and evacuation both via the access road and via the garden to the rear becomes difficult or even impossible for users.

For sensitive infrastructures, we recommend going beyond the hundred-year event in any case, both in terms of riverine and pluvial flooding. In this case, however, the infrastructure is not only exposed in the event of an extreme event, but already in the event of a hundred-year event or even less. Secondary effects on the Levana School, e.g. due to the loss of the power supply, are also very likely. However, the 2021 event flooded the school to a height of approx. 2 m, so that no electricity or water supply etc. could function in the building itself anyway. As the pupils are not permanently in the school building - as is the case in a nursing home, for example - any power-dependent supply devices must be operable on a mobile basis, at least for



a certain period of time. In the event of an incident, it is also not planned for the pupils to remain in the school permanently, but to evacuate sooner or later, so that the aspect of indirect exposure is not considered further. However, if a decision is made in favor of a climate-resilient (re)construction on site for a sensitive infrastructure, it is also necessary to examine any secondary effects, e.g. due to the failure of the power or water supply.

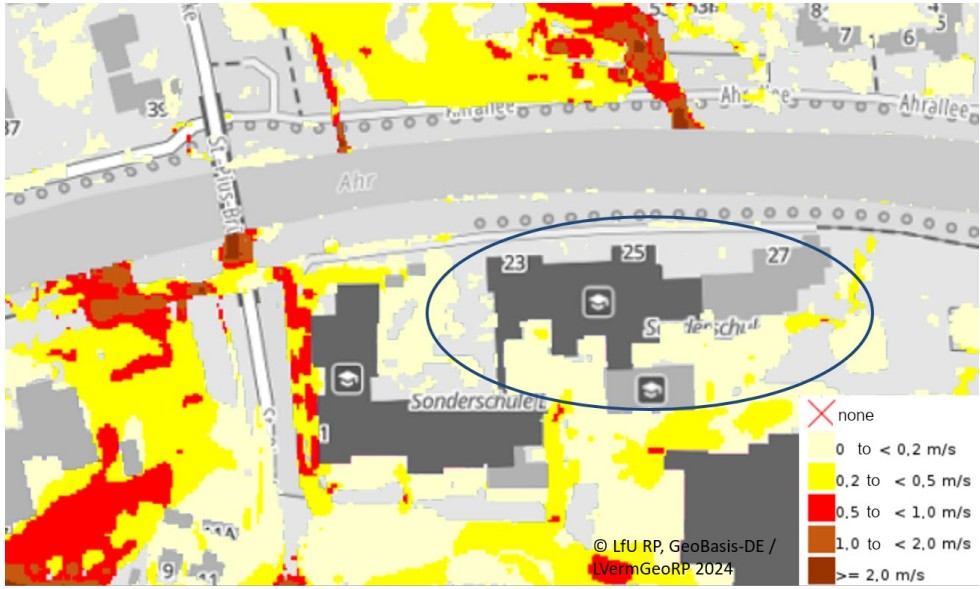

**Figure 4:** Flash flood hazard map in the vicinity of the Levana School with depiction of flow velocities (Sturzflutgefahrenkarte, Landesamt für Umwelt Rheinland-Pfalz (LfU RP); © GeoBasis-DE / LVermGeoRP 2024, https://wasserportal.rlp-umwelt.de/auskunftssysteme/sturzflutgefahrenkarten/sturzflutkarte (12.04.2024))

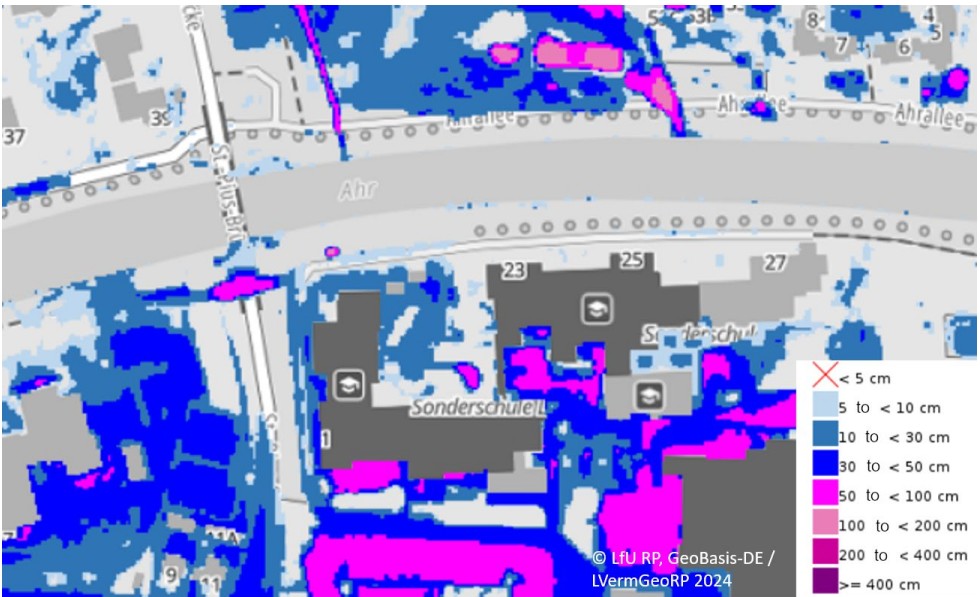

**Figure 5:** Flash flood hazard map in the vicinity of Levana School with depiction of flooding heights (Sturzflutgefahrenkarte, Landesamt 390 für Umwelt Rheinland-Pfalz (LfU RP); © GeoBasis-DE / LVermGeoRP 2024, https://wasserportal.rlp-umwelt.de/auskunftssysteme/sturzflutgefahrenkarten/sturzflutkarte (12.04.2024))



### 5.2.2 Vulnerability of the users

The students of the Levana School clearly belong to the most vulnerable groups. On the one hand, because of the limited mobility of an average of about 30 students (possibly even more in the future), which prevents them from getting to safety

independently and quickly, and on the other hand, because of the mental disabilities of all the students, which also make it difficult or even impossible for them to get to safety independently and quickly (see also Chapter. 5.2.4) - in the worst case, the students even put themselves in danger as the floods of 2016 had already shown (Former school staff member of Levana School, personal communication, 25.04.2023). As the pupils are very vulnerable due to their limited mobility and/or perception and need the help of third parties to evacuate, the Levana School can be identified as a sensitive infrastructure.

Furthermore, in this case and with regard to the specific issue of rebuilding such a school after an extreme flood event, it is also important to consider that, according to the assessment of a medical expert, children with an intelligence impairment can show inexplicable behavioral abnormalities, partly with auto-aggressive and xeno-aggressive behavioral disorders, when triggered e.g. by flowing noises (Ahr during floods or also during heavy rain), which can lead to a danger for fellow students, teachers and the students themselves (Medical Expert, personal communication, 2023). In addition, within this group of

persons, it can be assumed that the psychological recovery after a flood event is much more difficult and protracted compared to other groups of persons. For example, children with intelligence impairment (including learning disabilities) are difficult to therapize after trauma due to IQ and language impairment and as yet there are hardly any specialized diagnostic and therapeutic methods for this group of people (Mayer, 2020). Moreover, children with profound developmental disabilities require continuity of the learning environment (School staff member of Levana School, personal interview, 12.06.2023). Therefore,

changing the learning site again after another extreme event that cannot be ruled out and changing group assignments should be avoided if possible. If, as a result of another flooding event, schooling has to be carried out in a different location again, this will be very detrimental for these children, as they may be thrown back into unfavorable behavior patterns.

In addition to the extremely high vulnerability of the students, the school inventory of the Levana School is also very special and expensive, so that a renewed procurement of, for example, lifting platforms, swimming pool technology, teaching material

and rollators, seating devices, etc. after a loss due to another flood will be complicated, lengthy and expensive. During the flood of July 2021, helpers tried to secure the inventory until late at night, however, they put themselves in danger by doing so and were only able to move a small amount of inventory to safety (Former school staff member of Levana School, personal communication, 25.04.2023), as vertical relocation of the inventory was not possible due to the single-story construction of the building, leaving only the option of moving the inventory away with vehicles.

### 5.2.3 Vertical evacuation

In the case of Levana School, vertical evacuation is not possible as the school building is single-story due to its accessibility. Also, an evacuation to the roof has not yet been structurally provided for and is also not recommended due to the mental and physical limitations of the pupils. In addition, in extreme cases, as the 2021 event showed, the water can be several meters high, meaning that the roof of a single-story school building could also be flooded in future extreme events. This means that

this step of our assessment framework can be quickly ticked off with a "no", which leads to the final assessment step of "evacuation capability".

### 5.2.4 Evacuation capability

*Warning time:* An exact warning time for the evacuation of Levana School cannot be given for either a flood or a heavy rainfall scenario. Floods caused by heavy rainfall in particular often involve shorter warning times and higher uncertainty (Bronstert

et al., 2017). However, forecasts are always associated with uncertainties - in the summer of 2021, for example, the water level forecasts often only corresponded to the current water levels, which therefore did not include a longer warning time. The 2021 floods also showed that there are considerable problems with forecasting. For example, the water level forecasts were far too





low for a long time and later in the course of the event were congruent with the actual water level at the gauge in Altenahr (County of Ahrweiler, personal communication, 11.07.2023). The forecast therefore had no predictive effect for preliminary planning, e.g. for an evacuation. Also, higher water levels were measured in the upper reaches of the Ahr at midday, but effective early warning and evacuation did not take place (Weidinger, 2023). The later order, which was issued shortly before midnight, to evacuate entire settlement areas within 50 meters of the Ahr (Weidinger, 2023) also shows the lack of systematic evacuation and early warning as 50 meters was far too little. Even if it generally takes several hours for a flood from the upper Ahr Valley to reach Bad Neuenahr-Ahrweiler, it should be noted, that also larger tributaries of the Ahr, such as the Sahrbach, can lead to higher water levels in the Ahr and cause flooding, so that the advance warning time should not only be discussed based on the flood development in the upper Ahr Valley, as floods can also arise from inflows into the Ahr and the possible advance warning time can therefore be significantly shorter (SGD Nord, personal communication, 06.07.2023).

Although the exact warning times cannot be precisely defined from previous studies, it can be assumed that even with a warning time of several hours, there are considerable challenges in evacuating people and securing and relocating the specific inventory of the Levana School (Former school staff member of Levana School, personal communication, 25.04.2023; School staff member Levana School, personal interview, 12.06.2023) - as shown in the following.

*Evacuation time and condition of the pupils:* The Levana School has an evacuation plan, whereby the current evacuation plan, which we have been given access to, is for the container complex at the replacement location. The current evacuation time is given as eight to 10 minutes (fire department response time), whereby the evacuation only refers to leaving the building and not to the further evacuation from the flood risk area. This means that the existing evacuation plan is designed for the event of a fire; there is currently neither an evacuation plan nor comprehensive emergency exercises for the event of flooding.

In case of flooding, it must be taken into account that the collection point and access routes will already be flooded very quickly (before water enters the building). The time that is set at eight to 10 minutes for the simple evacuation of the building in the event of a fire will also be significantly longer in the event of heavy rainfall or flooding, as collection areas and access roads will also be affected and the entire site will have to be evacuated. Also, supervision by unknown teachers in the event of a flood disaster can lead to significant problems, up to complete refusal on the part of the child (School staff member of Levana School, personal interview, 2023). This can also occur if the child feels the panic and fear of the teachers. Even two years after the flood, some teachers as well as students are still afraid when it rains, and this feeling could be intensified in the old building (School staff member of Levana School, personal interview, 12.06.2023). In addition, it should also be noted that the particularly vulnerable groups accommodated at Levana School are in an exceptional situation due to their mental and physical limitations in the event of an evacuation, i.e. it is likely that some students will become unpredictable and will not stay in one location for a long time, but will often require individual support (School staff member of Levana School, personal interview, 12.06.2023). Furthermore, according to the expert opinion, many of the students perceive emotional states very accurately and thus sense the teachers' hectic, panic and anxiety, even if they try to suppress such feelings and states. The students, in turn, often mirror these emotional states, which makes evacuation, e.g. buckling up in the vehicle, waiting or being carried out, more difficult (School staff member of Levana School, personal interview, 12.06.2023). In addition to being affected at school, teachers are often also affected at home as well as in the family and worried about them (around 30% of the teaching staff were affected in 2021).

*Challenges in transportation and evacuation of the entire site:* Due to the single-story design (see Chapter 5.2.3), a vertical evacuation is not possible - therefore pupils and teachers will be forced to leave the school building and grounds in the event of flooding. As there are numerous children with physical and/or mental disabilities (currently at least 30 children with significant motor disabilities and a total of 92 children with disabilities, whereby the proportion of children with motor disabilities is likely to increase), such an evacuation to a flood-proof accommodation option outside the flooded area would require a considerable amount of time, personnel and technology. Many of the students are unable to leave the premises independently. Some of the older children, for example, would have to be carried by four people (School staff member of



Levana School, personal interview, 12.06.2023), and in rare cases pupils would even need a complete transport vehicle for themselves in order to be transported lying down out of the potential flooding area. As shown in Chapter 5.2.1, the building is already flooded by approx. 60 cm at an HQ-100 and parts of the access routes to the school are flooded even higher (see Fig. 3), so that safe accessibility is no longer guaranteed even well below the new HQ-100. Emergency ambulances according to

"Federal Office of Civil Protection and Disaster Assistance" (*Bundesamt für Bevölkerungsschutz und Katastrophenhilfe - BBK*) standards, which could be used for an evacuation, can drive through a maximum water depth of 30 cm (Bundesamt für Bevölkerungsschutz und Katastrophenhilfe, 2010) - similar values apply to other transporters, so that accessibility to the school can no longer be guaranteed in the event of an HQ-100, as parts of the access road are flooded by almost 1 m in this scenario (see Fig. 3). Even at shallower water depths than 30 cm, damage to the vehicle cannot be ruled out, as a bow wave is created,

for example, if the vehicle is driven through too quickly, which can damage the engine even at low water levels. A vehicle drifting in the water without a functioning engine is extremely dangerous, because in this case the water decides where the vehicle drifts. Another problem is the availability of such vehicles in the event of an incident. In principle, the vehicles of the school transport service are necessary for the evacuation of pupils away from the entire site, but this is a private service provider, which therefore does not have to be available immediately and in sufficient numbers in the event of an incident.

Corresponding emergency vehicles such as emergency ambulances could also be used during the evacuation, but these in turn are heavily involved during a widespread flood - on the one hand in the acute rescue of lives and on the other hand, in addition to the Levana School, around three kindergartens/day nurseries, four schools and two clinics are to be evacuated as further sensitive infrastructures within a radius of around one kilometer.

Alternatively, it would also be possible to evacuate the pupils via the garden, but this would again increase the distance

significantly - and it can be assumed that these paths are difficult to walk on in heavy rain and that self-evacuation here is out of the question, especially for children with limited motor skills who need a wheelchair, for example. Moreover, this type of evacuation would require a lot of staff, as some pupils can only be carried in fours - according to the school staff member, it is practically impossible to get to elevated and therefore to flood-safe areas in this way, i.e. without the buses (School staff member of Levana School, personal interview, 12.06.2023). In the area in front of the main entrance of the Levana School,

the water is about 60 cm high at an HQ-100, in the collection area even between 0.5 to 2 m high. This means, as well as transport vehicles, children can also no longer cross these areas at such water depths and higher flow velocities. According to studies, weaker people, including children in general and people with motor disabilities, are already swept over at a water depth of 20 cm (ankle-deep) and a flow velocity of around 3 m/s (MunichRE, 2016). According to a working aid published in 2019 by a Bavarian ministry, personal safety is no longer guaranteed even at 2 m/s and 20 cm (Bayerisches Staatsministerium

für Umwelt und Verbraucherschutz, 2019). For comparison: On the night of the flood, the flow velocity of the Ahr was several meters per second (dpa Rheinland-Pfalz/Saarland, 2022), according to estimates around 3-5 m/s (RedaktionsNetzwerk Deutschland, 2022) – and in the built-up areas up to 2 m/s could have been plausible (Apel et al., 2022). On top, there are also strong, unpredictable currents due to blockages and flow obstacles as well as floating debris, e.g. in the form of cars and trees, which can also sweep people away. Furthermore, for example, gully covers that have been washed up and exposed channel

inlets, which can develop a strong suction effect, can also become a deadly danger (Landesfeuerwehrschule Baden-Württemberg, 2023). Images taken after the flood event in 2021 also show that a high proportion of floating debris was washed up in this area. Particularly behind the St. Pius Bridge, which initially intercepts alluvial material, the very high flow velocity and strong current can lead to major damage if these accumulated masses are suddenly released. In addition to the water level and flow velocity, the destructive force and danger of debris must also be taken into account.

*Problems with remaining in the building:* It is also wrong to assume that pupils can simply remain at school in the event of a flood. Although the flood risk analysis only shows a slight flooding of the ground floor of the school in the event of an HQ-100, which can in principal be averted by structural measures, significantly higher flooding can also occur, as 2021 has shown.



In addition, flooding can also raise the floor slab, so that not only the flooding of the ground floor, but also the possible raising or partial destruction of the floor slab could pose a significant problem for the Levana School.

**5.2.5 Conclusion on the Levana School case study**

All in all, based on our assessment framework, the risk that the Levana School and its users are facing with regard to riverine and pluvial flooding is considered to be very high. Due to this very high risk, we strongly recommend the complete relocation of the school within the reconstruction process. Apart from the risk assessment, other factors also play a role in such a decision, e.g. financing or proximity to users and other facilities.

*Financing/Funding:* In terms of risk-based and resilient reconstruction and due to the extensive damage, that has already occurred, it is possible to finance the reconstruction of the school elsewhere from the reconstruction fund (Ministry of Interior and for Sports Rhineland-Palatinate, personal communication, 07.05.2024). Particularly with regard to sensitive infrastructures with high to very high risk, our assessment framework can also be used to qualify and support the decision to relocate.

*Proximity to users and other facilities:* In the case of the Levana School, this point can also be disregarded in principle as long

as relocation takes place within the county, as the school's catchment area covers the entire county. For other sensitive infrastructures such as kindergartens, the situation can be quite different. For example, our household survey showed that proximity to the facility is a very important criterion when choosing a facility (see Fig. 6).

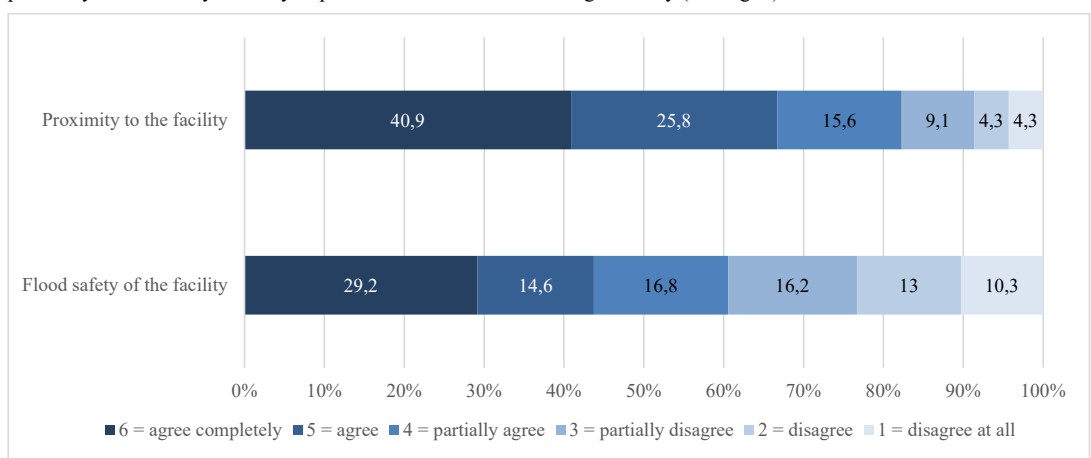

**Figure 6:** Agreement with the statement that "proximity to the facility" (n = 186) and "flood safety of the facility" (n = 185) are important
criteria when choosing a facility (hospital, nursing home, retirement home, kindergarten/daycare center, school); source: own household survey.

However, our survey also showed that the majority of respondents, i.e. over 80%, agreed with the statement that sensitive and critical infrastructure should be relocated away from the immediate vicinity of the Ahr (Truedinger et al., 2023). Hence, public acceptance for the relocation of such infrastructure after disasters is definitely present. And especially for infrastructures like
the Levana School, which serves the entire Ahrweiler county, it is likely also in the interest of the majority of users. Nevertheless, the availability of alternative locations is essential for relocation - and these locations must also meet certain criteria and requirements. For example, the pupils at Levana School also need certain everyday infrastructures such as supermarkets or road crossings nearby in order to be able to practice everyday life (School staff member, personal interview, 12.06.2023). Therefore, the location factor should not be disregarded. Nevertheless, as already mentioned, the Levana School
has a large catchment area and suitable alternative locations should therefore be found - so we strongly recommend relocating the school and building a climate-resilient new school in a different, safer location, as the risk could be classified as very high using our assessment method.



## 6 Discussion

There are various ways of increasing resilience to flooding and heavy rainfall in the construction and reconstruction of sensitive

infrastructure. Within this process, our assessment framework can be used to qualify and accelerate (funding) decisions regarding the type of reconstruction and possible relocation.

### 6.1 Options for strengthening the resilience of sensitive infrastructures during reconstruction

Depending on the expected risk to the sensitive infrastructure and the factors that have contributed to this specific risk assessment, an appropriate approach can be taken. Regularly reassessing the risk evaluation is generally advisable, as factors

may change over time due to, e.g., climate change or shifts in user demographic. In addition to the risk, other factors always come into play, such as the availability of financial resources. For the reconstruction in the Ahr Valley and the other affected regions, financial resources amounting to billions of euros have been made available by the federal and state governments; however, the use of these funds is subject to administrative regulations and thus state requirements. For example, in some cases, the relocation of infrastructure cannot be fully financed through the reconstruction fund, as it primarily concerns damage

compensation (Birkmann et al., 2023).

If an infrastructure is not exposed in any way, no further measures are required in principle. However, exposure can change, e.g. due to climate change, so regular checks are advisable. In addition, resilience-enhancing measures can still be considered if the appropriate personnel and financial capacities are available, e.g. regular evacuation exercises to the roof or away from the entire site or the known structural protection measures (Brombach et al., 2013) - such as pressure-tight windows or

waterproof materials.

If an infrastructure is directly exposed, but the users are not particularly vulnerable and do not require third-party assistance in the event of an incident, the infrastructure could not be classified as sensitive infrastructure - the known structural and organizational protective measures are nevertheless recommended to protect the building and its functions from damage.

If a vertical evacuation to a vertical height that is high enough to guarantee safety for all the hazard levels, even for rare extreme

events, is possible in a short time and with the personnel available on a daily basis, which can be checked regularly through appropriate exercises, the known structural and organizational protective measures are also recommended (Brombach et al., 2013). In addition, regular evacuation exercises and revisions of the evacuation plans should be carried out, whereby consideration should also be given to how evacuation from the roof can be carried out in the later course of an incident. Moreover, the immediate and accessible information of all users about an evacuation to the roof must be ensured.

If evacuation capability is not given and/or vertical evacuation is not or only insufficiently possible, there is a high to very high risk, and relocation of the site is strongly recommended. Of course, other factors also play a role in site selection, such as the distance to users (see Fig. 6) or other facilities, e.g., to practice everyday; however, in the case of high to very high risk, a high priority should be given to flood safety.

In the case of indirect exposure, measures should also be taken depending on the extent and nature of the impact. For example,

if indirect impacts due to a power outage are possible, a redundant supply can be implemented through a second local substation located outside the flood-prone area. Additionally, emergency power supply can be installed on higher floors or external power feed-in options can be established. If the risk cannot be reduced through such resilience-enhancing measures, consideration must also be given to complete relocation in the case of high to very high risk.

The approach developed and tested here can therefore be used as a justification for flood-adapted reconstruction or relocation

of sensitive infrastructures as part of the reconstruction process. From the perspective of urban land-use planning, the approach can also be applied to future projects in order to either evaluate sites or alternative sites in terms of their resilience or to set a framework for flood-adapted construction methods.



### 6.2 Transferability, applicability and benefits of the assessment framework

As demonstrated by the case study of the Levana School which is a special school with a focus on mental as well as motor and
physical development with a highly vulnerable user group, our assessment framework can be effectively applied to determine
the risk that a sensitive infrastructure is facing with regard to heavy rain and flooding. Building upon this, appropriate measures
can then be derived, potentially qualifying and even expediting reconstruction planning as the assessment framework also
enables prioritization of measures for various infrastructures in the reconstruction process. And qualifying in the sense that, in
the context of climate-resilient and sustainable planning and reconstruction, relocation should be financially and, if possible,
technically supported, particularly in cases of high to very high risk, by government entities. Accordingly, funding regulations
should also be adjusted to enable financing beyond the mere extent of the damages, if, for example, this is warranted due to
the high risk and the protection worthiness of the infrastructure which can be determined by such new assessment frameworks.
Reconstruction should also be pursued with the aim of climate adaptation and resilience and the economical use of tax
resources, so that modification and moderate settlement withdrawal are indicated, especially for particularly sensitive
infrastructures that are worthy of protection, and the window of opportunity for reconstruction is used optimally (Birkmann et
al., 2023).

The applicability of the assessment framework extends to other infrastructures such as nursing homes or kindergartens, with
the Levana School serving as a transferable example. However, there are limitations to its application, such as data availability.
Without sufficient data on flood and heavy rain hazards, user groups, or evacuation capability, a comprehensive risk assessment
cannot be conducted. Additionally, it should be noted that during a real disaster, processes may not necessarily unfold as
planned and practiced beforehand. Therefore, we recommend higher standards of protection for sensitive infrastructures
(Birkmann et al., 2022) and, where possible, relocation to less exposed areas.

### 7 Conclusions

Our paper has highlighted the current legal requirements and definitions regarding critical and sensitive infrastructure in
Germany - and what the difference between critical and sensitive infrastructures is. We understand sensitive infrastructures as
those that are not necessarily essential for the functioning of society but can still be of significant importance and host
particularly vulnerable user groups who may require assistance from third parties in case of an incident. Due to the lack of
procedures and approaches for identifying such sensitive infrastructures, assessing their risk, and developing resilience-
enhancing measures, we have developed a new assessment framework that allows for the identification of sensitive
infrastructure and the assessment of risk. Furthermore, through a literature review, expert interviews, and investigations into
the specific reconstruction process, we were able to identify corresponding measures that can be taken to enhance resilience
in the reconstruction process depending on the risk. Inhibiting factors, especially the financing of such measures like relocation,
were also identified. We recommend revising funding guidelines or adjusting future funding guidelines to enable greater
resilience in reconstruction. Especially in the case of high to very high risk, it is recommended to consider and, if possible,
implement relocation of sensitive infrastructures in terms of the protection worthiness. Ultimately, it is always a matter of
weighing up the various factors such as the flood risk, the location with proximity to certain other infrastructures, the financing,
the structural and organizational precautionary options, the demands placed on the infrastructure by the users and so on.
However, greater emphasis should be placed on risk, especially in reconstruction, when it comes to sensitive infrastructures
which is enabled by our assessment framework. Due to the high protection worthiness, climate change, and the fact that
sensitive infrastructures are often intended to provide a safe location for decades, we also recommend that when considering
exposure and relocation not to use events with a statistical return period of one hundred years as the design events, but rather
extreme events. Especially in reconstruction, the assessment framework can also be used to prioritize, thereby accelerating
processes and qualifying decisions, such as those regarding site selection and the respective funding. Furthermore, it can ensure



that the needs of particularly vulnerable population groups can be taken into account more strongly and systematically in
reconstruction and new construction.

Since our focus was on the assessment framework, which we also verified using a real-life example, future research could
focus more in-depth on the resulting resilience measures. Moreover, the assessment framework could also be extended or
generalized with regard to other natural hazards in the future. Additionally, an adaptation of the framework for critical
infrastructure could also be investigated.

**Data availability**

The survey data used can be found directly in the manuscript. Some of the confidential minutes and personal communication
cannot be made public due to data protection. However, mainly publicly accessible sources were used. If specific data is
required, please contact the corresponding author.

**Author contribution**

AT and JB conceptualized the paper and developed the methodology. AT conducted the investigation with partial support from
JB and MF. AT prepared the original draft including visualization and JB, MF and CF reviewed and edited the manuscript.

**Competing interests**

The authors declare that they have no conflict of interest.

**Acknowledgements**

We are very grateful for the funding provided by the Federal Ministry of Education and Research/Bundesministerium für
Bildung und Forschung (BMBF) as part of the KAHR project, grant number (FKZ) 01LR2102A.

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
