# Peer review of "Sensitive infrastructures and people with disabilities – Key issues when strengthening resilience in reconstruction"

_EGUsphere, 2024_

## Author Comment (AC1)

**RC1**

Dear authors,

Firstly, thank you very much for your submission of your paper to the NHESS journal. The paper addresses a very interesting and important topic, which warrants further in-depth consideration within the field of vulnerability research. I have some general comments, which I would like to address below:

Dear Reviewer 1,

thank you very much for the kind and very helpful review.

Overall, I would like to ask you to reconsider four core issues:

a) The current version lacks a strong theoretical framework. Section 3 tries to initiate a debate on why we need to rethink the current critical infrastructure literature. I suggest shifting some parts of section 3 to section 1, or creating a new section, where you provide a broader theoretical discourse.

Thank you, that is an important point. We will reconsider shifting some parts from section 3 to section 1. Moreover, we will definitely add some more theoretical background, especially with regard to the risk framework of the Intergovernmental Panel on Climate Change (IPCC) and to the components that risk is made up of within this risk concept, since it is on this theoretical basis (exposure and vulnerability, including coping and adaptive capacity) that our framework for risk assessment for sensitive infrastructure has been developed. We will also refer to further publications that have dealt with specific indicators for risk assessment.

b) The lack of a broader theoretical discussion also affects your discussion section, which does not read like a discussion. You need to show how your results/paper link to the ongoing literature and what your new theoretical contribution is.

Thank you. We would be happy to revise this as well, particularly with regard to the additions to the theoretical background already mentioned under a). Our new contribution here is the concrete application of the theoretical, established risk framework to sensitive infrastructures and their user groups and the comprehensible and easy applicability.

c) The research questions need to be reconsidered. Provide a more in-depth explanation of why these questions are important and what the new and innovative contribution of the paper is.

Thank you. We will also reconsider the research questions and elaborate the questions more specifically in relation to the content of our paper and the new findings - an easy-to-use framework for assessing the risk of sensitive infrastructure and the resulting implications.

d) The method section needs to be reconsidered. The literature review in your paper does not need to be mentioned, as it does not employ a proper scientific meta-analysis or systematic literature review. Furthermore, regarding the household survey, I am unsure of its added value.

Clarify what you aim to answer with the survey. It might be beneficial to rethink which methods and data you can include in the paper to address your research questions and the gap in the literature.

Thank you. We will remove the literature review as it hasn't been a systematic literature review. Moreover, we will look into our household survey data to see to what extent the survey data can generate added value for our paper and how the survey results can meaningfully underline the content.

A minor aspect: Sections 3.2 and 3.3 are interesting but feel somewhat misplaced in the paper. While I see the value of both sections, they might need to be relocated to different parts of the paper. Regarding Section 5.2.5: first, I am not sure if the title is appropriate, and second, it currently only uses descriptive statistics. More analysis is needed to justify the inclusion of this data in your paper.

Thank you. We will try to relocate those both sections (3.2 and 3.3), but we would not want to delete them, as they are important for the context of the paper. The title of section 5.2.5 is a bit misleading, since it is not exclusively about a conclusion regarding the Levana School, but also considers other aspects – we are therefore happy to adjust the title.

---

## Author Comment (AC2)

**RC2**

*Dear Authors,*

*Thank you for submitting your manuscript and the extensive work it represents. Your research offers a valuable contribution to understanding the role of sensitive infrastructures, particularly those serving vulnerable populations. The focus on disabled communities and sensitive infrastructures is timely and highlights an often-underexplored area in disaster risk management.*

Dear Reviewer 2,

thank you so much for your valuable and constructive review.

*Some clarification is needed on the following key points:*

*The paper currently positions itself as a mixed-methods study, but the primary approach is qualitative, with reliance on interviews and document analysis. To ensure clarity and methodological coherence, it would be more accurate to frame the study as qualitative. If quantitative elements are present, they should be better integrated and explicitly discussed throughout the manuscript. I don't see the integration of the household survey, or its benefit for the current manuscript.*

Thank you for this helpful comment. We agree that it makes more sense, given the large qualitative share, to specify a qualitative approach as the method. Since the other reviewer also had comments on the integration of the household survey, we will take another critical look at the data from our survey and either remove these data completely or integrate the household survey data better and in more detail.

*The paper mentions the use of expert interviews and workshops as part of the research, but the details of these qualitative methods are underdeveloped. To strengthen the methodological rigor, it would be helpful to provide more information on how the interviews and workshops were conducted (e.g., participant selection, structure of interviews or workshops, the specific data analysis techniques). Additionally, it is important to address any potential biases in the data collection process, and how these were mitigated. A description of how the qualitative data were evaluated, whether through thematic analysis, coding, or another method, would also be valuable.*

Thank you, that is also a good point. We can specify this further with regard to the mentioned points, as this will improve the methods section and the scientific value of our paper.

*There is strong emphasis on regulations and their implementation, but the theoretical foundations related to vulnerability, resilience, disaster risk management, coping, and adaptation could be more clearly articulated. Currently, these concepts appear in the text, but they are used somewhat interchangeably. A more distinct differentiation of these concepts would provide greater clarity and coherence. Additionally, the literature review is not well defined and currently not supported.*

Thank you. This comment also coincides with the other review, and we fully agree that the theoretical background was not yet sufficiently developed and presented, even though the IPCC's risk concept was the basis for our assessment framework. We are happy to provide more in-depth theoretical background and to refer to other studies that have already made risk measurable in other contexts through exposure and vulnerability (susceptibility, coping and adaptive capacity). The new

contribution on our part is that we have made the already established risk concept applicable and comprehensible to the specific case of sensitive infrastructures.

*While the paper rightly focuses on people with disabilities, their voices seem absent from the data. Were individuals with disabilities, or their representatives, included in the interviews or consultation processes? Including these perspectives directly would add depth to the findings and make the research more inclusive.*

Thank you. We were unable to include the students themselves for a number of reasons. On the one hand, the students are still minors, which is problematic from a data protection perspective. On the other hand, they were already severely distressed by the flood event, which is why making contact on the subject of the flood was difficult for ethical reasons. Nevertheless, we spoke to both teachers and the parent representative of the school who represents the interests of the student body.

*The paper uses exposure as the primary criterion within its novel framework, but this focus might be too narrow. The manuscript acknowledges limitations such as data issues, high uncertainty, and climate change, challenging hazard maps. Since hazard maps may not fully reflect current or future risks, particularly given the Ahrtal flood example, is the focus on physical exposure alone justified? The concept of the "levee effect" could further challenge this focus. Additionally, while the second decision level introduces vulnerability and coping, these concepts are not clearly defined or supported with sufficient theoretical discussion, leaving their role in the framework unclear.*

Thank you very much for this very helpful comment. We will consider this especially with regard to climate change, the ubiquitous occurrence of heavy rainfall events and possible "levee effects". On the other hand, we are not only arguing with the HQ-100, which is legally binding in Germany, but also with the HQ-extreme, which already includes extreme events. Furthermore, in the context of reconstruction, there is definitely a risk and an event has already occurred, so the starting point via the exposure, which has already been met in this case, is understandable from our point of view. In the affected regions, such as the Ahr Valley or Spain, the patterns of exposure are significant. Especially in the Ahr valley, the exposure is enormous and has become very visible. In addition, there are already many heavy rain maps in Germany with a large extent, in which potentially many areas could be affected. Therefore, the approach via exposure seemed quite reasonable and logical to us. Nevertheless, we are also convinced that research on exposure must go further, that extreme events in particular must be included, that scenarios such as the breaking of dikes and other protective measures must also be considered, and that the scenarios considered should therefore be defined more generally.

We will think again about a possible adjustment of the framework and whether, for example, a risk matrix would be a viable option. Furthermore, in the conclusions, we will clearly elaborate on the importance of adjustments with regard to the exposure scenarios used.

*The method and results sections feel somewhat disconnected, and the flow between them could be improved. Furthermore, the discussion introduces new perspectives that do not always clearly link back to the research questions, methods, or results. Reworking the structure to ensure that the findings align with the initial research questions would improve coherence.*

Thank you very much. Since we will be making some additions and restructuring anyway - both in response to your comments and to the comments of reviewer 1 - we will also bear this comment in mind.

*Minor points for clarification:*

*The maps provided are important, but their readability could be improved. Specifically, the resolution and color schemes may not be suitable for readers with color vision deficiencies. Ensuring that maps are color-blind friendly and enhancing the visibility of key features like roads and schools, and complete legends would be beneficial.*

Thank you – we will revise it to enhance visibility.

*The manuscript is somewhat lengthy and sometimes lacks a clear focus. Tightening the narrative and ensuring smoother transitions between sections would enhance readability. Guiding the reader more explicitly through the structure of the paper would help maintain engagement and improve clarity.*

*Improving the clarity of the terms used, especially when discussing theoretical concepts like vulnerability and resilience, would help avoid confusion and increase the precision of the argument.*

Thank you. We will also bear those last two comments in mind when revising our manuscript and tighten the narrative and improve the clarity.